# Newer and Emerging LDL-C Lowering Agents and Implications for ASCVD Residual Risk

**DOI:** 10.3390/jcm11154611

**Published:** 2022-08-08

**Authors:** Rishi Rikhi, Michael D. Shapiro

**Affiliations:** Center for Prevention of Cardiovascular Disease, Section of Cardiovascular Medicine, Department of Internal Medicine, Wake Forest School of Medicine, Winston-Salem, NC 27101, USA

**Keywords:** dyslipidemia, cardiovascular disease, prevention, treatment, risk factors

## Abstract

Multiple lines of evidence demonstrate that low-density lipoprotein-cholesterol causes atherosclerotic cardiovascular disease. Thus, targeting and lowering low-density lipoprotein-cholesterol is the principal strategy to reduce cardiovascular disease risk in primary and secondary prevention. Statin therapy is the foundation of lipid-lowering treatment, but adherence rates are low, and many individuals do not attain target low-density lipoprotein-cholesterol values. Additionally, most statin-treated patients are still at considerable atherosclerotic cardiovascular disease risk, emphasizing the need for more aggressive low-density lipoprotein-cholesterol-lowering therapies. The purpose of this review is to discuss new and emerging approaches to further lower low-density lipoprotein-cholesterol, including inhibition of ATP-citrate lyase, proprotein convertase subtilisin-kexin type 9, angiopoietin-related protein 3, and cholesteryl ester transfer protein.

## 1. Introduction

For the past three decades, statin therapy has been the cornerstone for reducing atherosclerotic cardiovascular disease (ASCVD) risk [1,2]. During this time, numerous studies have demonstrated an indisputable causal role between low-density lipoprotein-cholesterol (LDL-C) and ASCVD [2]. Thus, therapies that decrease cumulative lifetime exposure to LDL-C have been efficacious in lowering ASCVD risk [2,3]. However, despite the efficacy and cost-effectiveness of statin therapy, ASCVD remains the leading cause of death globally [4,5,6]. Importantly, many of those on statin therapy continue to have residual ASCVD risk and are unable to achieve target LDL-C goals [1].

More recently, newer non-statin lipid-lowering therapies have surfaced with high efficacy in lowering LDL-C. The purpose of this review is to discuss mainstay LDL-C-lowering treatments, as well as emerging therapies that show promise in further reducing ASCVD risk (Table 1).

## 2. Statins

Statins, originally derived from fungus, decrease LDL-C by inhibiting the rate-limiting enzyme, β-hydroxy β-methylglutaryl-coenzyme A (HMG)-CoA reductase, in the cholesterol synthesis pathway [30]. This results in increased expression of LDL receptors in the liver and, therefore, higher uptake of LDL-C from circulation, driving down plasma LDL-C [30]. Although discovered in the 1970s, it was not until 1987 when the first statin, lovastatin, became commercially available [30,31]. In 1997, statins gained significant notoriety as they were shown to decrease CVD risk (HR 0.70, 95% CI 0.58, 0.85) in the landmark Scandinavian Simvastatin Survival Study [32]. Subsequently, several more statin compounds have been commercialized, with different LDL-C-lowering efficacies [31].

Since their discovery, statins have been extensively studied in randomized clinical trials and are the first-line therapy in primary and secondary prevention of ASCVD [1,7]. The most important evidence for statin efficacy comes from the 2010 meta-analyses of 26 randomized controlled trials from the Cholesterol Treatment Trialists’ Collaboration that included individual data from 169,138 participants, showing that a reduction in LDL-C by approximately 39 mg/dL led to a 22% reduction in major vascular events over the span of 5 years, even when LDL-C was low [8]. Despite the efficacy and excellent side-effect profile, where myopathy is expected to occur in 1 out of every 10,000 patient-years, adherence rates are poor and many individuals on statin therapy develop ASCVD events [1,4,33]. Thus, non-statin lipid-lowering therapies are under investigation and currently in use to reduce residual risk in certain patient populations.

## 3. Ezetimibe

Cholesterol homeostasis in the body is governed by the liver and intestine, of which the latter is the target for ezetimibe, the most used non-statin lipid-lowering therapy [7,34]. Ezetimibe targets the Niemann-Pick C1-Like 1 (NPC1L1) transporter found in the jejunal brush border, decreasing cholesterol absorption by 50% [1,33,35]. By doing so, less cholesterol makes its way to the liver, resulting in increased expression of LDL receptors and, therefore, reduced plasma LDL-C, by approximately 20% [1,4]. The most convincing evidence of the cardiovascular benefit of adding ezetimibe to statin therapy comes from The Improved Reduction of Outcomes: Vytorin Efficacy International Trial (IMPROVE-IT) [36]. Here, 18,144 patients at least 50 years of age with recent acute coronary syndrome were randomly assigned to receive simvastatin 40 mg plus placebo daily (*n* = 9077) or simvastatin 40 mg plus ezetimibe 10 mg daily (*n* = 9067) [36]. After 7 years of follow up, the group with ezetimibe had decreased risk of the primary composite endpoint (HR 0.936, 95% CI 0.89, 0.99) and 24% lower LDL-C [36]. The decreased risk was notable early at approximately 1 year and seems to be driven predominately by reduced incidence of myocardial infarction and stroke [36]. These findings were confirmed in a large meta-analysis of 21,727 individuals, where the authors found reduced risk of major cardiovascular events with the addition of ezetimibe to statin therapy (HR 0.94, 95% CI 0.90, 0.98) [37]. It is worth mentioning that the cardiovascular risk reduction was primarily in those with established ASCVD and there was no benefit in fatal outcomes with the addition of ezetimibe [37].

These findings illustrate the cardiovascular benefit of LDL-C-lowering and how it is not intrinsic to only statin therapy [1,36]. Based on this evidence, ezetimibe is recommended as add-on therapy in secondary prevention for those who have an LDL-C ≥ 70 mg/dL while on the highest tolerated statin dose [7]. Additionally, ezetimibe can be added to maximally tolerated statin therapy in those with severe primary hypercholesterolemia who have an LDL-C ≥ 100 mg/dL [7]. Although effective in reducing cardiovascular risk, the added LDL-C lowering impact of ezetimibe is modest; thus, newer therapies to further drive down plasma LDL-C and cardiovascular risk are needed [4,36].

## 4. Bempedoic Acid

The non-statin lipid lowering prodrug, 8-hydroxy-2,2,14,14-tetramethylpentadecanedioic (bempedoic) acid, is metabolized in the liver and targets ATP-citrate lyase in the cholesterol synthesis pathway, upstream from HMG-CoA reductase [1,4,10]. As a prodrug, bempedoic acid requires enzymatic activation in the liver, minimizing off target effects, including skeletal muscle, and thus, may be of value in those who experience myopathy symptoms associated with statin use [1,4,10]. The safety and efficacy of bempedoic acid has been described in recent phase 3 Cholesterol Lowering via Bempedoic acid, an ACL-Inhibiting Regimen (CLEAR) trials [9,11,38,39].

Clear Serenity, a phase 3 trial, randomized 345 participants with statin intolerance and an LDL-C ≥ 130 mg/dL for primary prevention or an LDL-C ≥ 100 mg/dL in those with heterozygous familial hypercholesteremia or history of ASCVD, to receive bempedoic acid or placebo [39]. The study found that bempedoic acid was effective at reducing LDL-C by 21.4% and caused less myalgias than the placebo group at 12 weeks [39]. Even more pronounced LDL-C-lowering was seen in CLEAR Tranquility, which randomized 269 participants with an LDL-C ≥ 100 mg/dL on ezetimibe therapy to bempedoic acid versus placebo [38]. After 12 weeks, bempedoic acid was able to reduce LDL-C by 28.5% with similar adverse events compared to placebo [38]. More long-term follow up was completed in the CLEAR Harmony trial, where 2230 participants with a history of ASCVD and/or heterozygous familial hypercholesterolemia on maximally tolerated statin therapy with an LDL-C ≥ 70 mg/dL were randomized to bempedoic acid or placebo [11]. The bempedoic acid group had significantly lower LDL-C at 52 weeks, with the highest LDL-C reduction, 18.1%, compared to placebo, occurring at 12 weeks [11]. Additionally, adverse events in the bempedoic acid and placebo group were comparable at 52 weeks; although, the bempedoic acid group had higher uric acid levels and gout events [11]. In a very similar designed randomized trial, CLEAR Wisdom, 779 participants with an initial LDL-C ≥ 100 mg/dL were included [9]. After 12 weeks, the bempedoic acid group had an LDL-C reduction of 17.4% compared to the placebo that was largely sustained through 52 weeks [9]. A more recent randomized controlled trial investigated the use of a fixed-dose combination of bempedoic acid and ezetimibe in 301 participants with elevated CVD risk [40]. The group receiving the fixed-dose combination had an LDL-C reduction of 38.0% compared to the placebo at 12 weeks [40]. These trials indicate a promising role for bempedoic acid; although, cardiovascular outcome data are not yet available, but are currently being investigated in the fully recruited CLEAR Outcomes trial (ClinicalTrials.gov Identified NCT02993406) [40]. In February 2020, the FDA approved bempedoic acid and the fixed-dose combination with ezetimibe for use in those with a history of ASCVD or heterozygous familial hypercholesterolemia who are on maximally tolerated lipid-lowering therapy and in need of further LDL-C reduction [10].

## 5. Proprotein Convertase Subtilisin-Kexin Type 9 (PCSK9) Inhibition

Over the past 20 years, much has been learned about the PCSK9 pathway, stemming from the breakthrough finding in 2003 where gain of function mutations in *PCSK9* were shown to be the third locus of familial hypercholesterolemia [41,42]. PCSK9 is a serine protease secreted from hepatocytes into circulation and targets the extracellular surface of the LDL receptor, signaling it for lysosomal degradation [41,43]. This destruction of the LDL receptor results in decreased quantity of LDL receptors on the hepatic surface, and thus, less LDL particle clearance [41,43]. Moreover, loss of function in *PCSK9* is associated with significantly lower LDL-C and cardioprotection, highlighting the potential of PCSK9 as a therapeutic target [41,43,44].

One of the methods for inhibiting PCSK9 is with fully human monoclonal antibodies, of which alirocumab and evolocumab have been well studied in secondary prevention trials, Evaluation of Cardiovascular Outcomes After an Acute Coronary Syndrome During Treatment with Alirocumab (ODYSSEY OUTCOMES) and Further Cardiovascular Outcomes Research with PCSK9 Inhibition in Subjects with Elevated Risk (FOURIER), respectively, capable of lowering LDL-C by approximately 60% [1,45,46]. In ODYSSEY OUTCOMES, 18,924 individuals with recent acute coronary syndrome on maximally tolerated statin therapy were randomized to receive alirocumab versus placebo [45]. After a median of 2.8 years, the investigators found decreased risk in the primary composite outcome in the alirocumab group (HR 0.85, 95% CI 0.78, 0.93) [45]. A similar reduction in cardiovascular risk was seen in FOURIER, which randomized 27,564 participants with a history of ASCVD and elevated cardiovascular risk, on maximally tolerated statin therapy, to evolocumab versus placebo [47]. After a median of 2.2 years, the group treated with evolocumab had lower risk of the primary composite outcome (HR 0.85, 95% 0.79, 0.92) [47]. In both ODYSSEY OUTCOMES and FOURIER, therapy was well tolerated with no major differences in adverse events between treatment versus placebo groups, despite dramatic reductions in LDL-C [45,47]. These trial results highlight the LDL-C-lowering potential and cardioprotective impact of alirocumab and evolocumab; however, these medications are currently much more expensive than ezetimibe and statin therapy, at approximately USD $6000 a year [45,46,47]. Thus, current recommendations are to use alirocumab and evolocumab as add-on therapy to maximally tolerated statin and ezetimibe treatment, when LDL-C is ≥70 mg/dL in secondary prevention or ≥100 mg/dL for those with severe primary hypercholesterolemia [7].

Another method to target PCSK9 is with inclisiran, a small interfering RNA (siRNA) therapy that inhibits translation of messenger-RNA (mRNA) [48,49]. Inclisiran has two strands, each 21–23 nucleotides in length, that interacts with the RNA-induced silencing complex (RISC), which then binds with PCSK9 mRNA, preventing translation of the PCSK9 protein [48,49]. During drug development, several chemical modifications occur to increase the compound’s potency and duration of action [48,49]. Additionally, N-acetylgalactosamine is added to the sense strand, targeting inclisiran to the liver, minimizing systemic adverse effects [48,49]. Although the half-life of inclisiran is only 9 h, the guide strand and RISC remain active and capable of interacting with PCSK9 mRNA multiple times, prolonging the clinical efficacy duration [48,49]. Much of the pharmacology and efficacy surrounding inclisiran comes from the phase 1, 2, and 3 ORION clinical trials [50].

ORION-1 randomly assigned 501 participants with an LDL ≥ 70 mg/dL and a history of ASCVD or LDL-C ≥ 100 mg/dL and no history of ASCVD to inclisiran versus placebo in a multiple-ascending dose phase 2 trial [51]. The authors found a significant and sustained reduction in LDL-C at 6 months in those who received one dose of inclisiran compared to placebo (30–44%) [51]. This reduction was even higher in those who received two injections of inclisiran, at day 1 and day 90, compared to placebo (37.3–54.4%), signifying a dose-dependent response [51]. This profound and sustained reduction in LDL-C was also seen in three phase 3 inclisiran trials, ORION-9, -10 and -11, all of which included patients with elevated cardiovascular risk, with LDL-C not at the goal on maximally tolerated statin therapy [52]. Additionally, the intervention group in all three trials received inclisiran sodium (300 mg) at day 1, 90, and every six months thereafter [52,53]. In ORION-10 and -11, at 510 days, LDL-C was 52.3% and 49.9% lower in the inclisiran group, respectively, compared to placebo [52]. Although long term safety data are not available, the analysis of the ORION-10 and -11 trials included data on a total of 2166 person-years, which did not show significant differences in adverse events between inclisiran and placebo groups [52]. Similar results were seen in ORION-9, which included 482 individuals with heterozygous familial hypercholesterolemia [53]. After 510 days, the inclisiran group had a reduction in LDL-C of 47.9%, with similar adverse events, compared to placebo [53]. In a recent meta-analysis of 3660 participants from ORION-9, -10, and -11, the authors found that inclisiran reduces LDL-C by 51% and major adverse cardiac events by 24%; however, these trials were not powered to assess clinical outcomes [12]. Fortunately, several other inclisiran clinical trials are underway, with the ORION-4 trial assessing major adverse cardiac event outcomes (ClinicalTrials.gov Identified NCT03705234) [50]. Based on the available evidence, the FDA approved use of inclisiran in the United States on 22 December 2021 as additional lipid-lowering therapy for those with a history of ASCVD or heterozygous familial hypercholesterolemia [13].

An emerging method of targeting PCSK9 is with oral therapy, avoiding the need for injections. MK-0616 is a macrocyclic peptide that was recently investigated in two phase 1 studies and found to lower LDL-C by 65%, with no significant adverse events [14,15]. Given these promising phase 1 results, there are plans for a phase 2 trial later this year [16]. Another oral therapy in development is a highly potent ASO, AZD8233, which was found to reduce LDL-C by 45–50% in cynomolgus monkeys and was mostly well tolerated [17].

A permanent approach to gene silencing is underway with Clustered Regularly Interspaced Short Palindromic Repeat (CRISPR) base editing of *PCSK9* [18]. Recently, *PCSK9* loss of function mutation was delivered via CRISPR adenine base editing therapy, VERVE-101, to non-human primates, resulting in a 90% reduction in circulating PCSK9 and 60% lowering of LDL-C [18]. This type of technology offers a paradigm shift in cholesterol management, omitting oral medications and frequent injections; however, long-term studies will be needed to assess the safety of permanent silencing of *PCSK9* and possible off target effects [18].

## 6. Angiopoietin-Related Protein 3 (ANGPTL3)

ANGPTL3 is a polypeptide hepatokine that inhibits lipoprotein lipase, an enzyme involved in the hydrolysis of triglycerides that plays a key role in the metabolism of very-low-density lipoprotein-cholesterol (VLDL-C) to LDL-C [54]. Additionally, ANGPTL3 inhibits endothelial lipase, a phospholipase pivotal to HDL-C metabolism [54]. There has been significant interest in the role of ANGPTL3 in the lipid metabolism pathway, as individuals with loss of function of *ANGPTL3* have hypobetalipoproteinemia, including 70% lower LDL-C levels [21]. While the mechanism of lowering LDL-C is not completely understood, it is thought that blocking ANGPTL3 increases the clearance of remnant lipoproteins, preventing the formation of LDL-C [54].

One method of blocking ANGPTL3 is with evinacumab, a fully human monoclonal antibody that was shown in a phase 1 study of 83 healthy volunteers to reduce LDL-C by 23.2% [55]. In a phase 2 trial involving nine participants with homozygous familial hypercholesterolemia with a mean LDL-C of 376 mg/dL, evinacumab was able to reduce LDL-C by 49% after 1 month [56]. Similar results were seen in another phase 2 randomized controlled trial of 272 participants with primary hypercholesterolemia on maximally tolerated lipid lowering therapy, where LDL-C was reduced by 56% and 50.5% by subcutaneous and intravenous evinacumab at 16 weeks, respectively [57]. More recently, a phase 3 trial randomized 65 patients with homozygous familial hypercholesterolemia on maximally tolerated lipid lowering therapy with a mean LDL-C of 255.1 mg/dL to evinacumab versus placebo [19]. The study found a decrease in LDL-C of 49% compared to placebo at 24 weeks with no major differences in adverse events [19]. While safety data from these trials are promising, long-term safety data of evinacumab are unknown, but will be addressed in a fully recruited phase 3 trial (ClinicalTrials.gov Identified NCT03409744) [58]. In 2021, the FDA approved evinacumab for individuals with homozygous familial hypercholesterolemia and at least 12 years of age [20].

ANGPTL3 can also be targeted via gene-silencing mechanisms, including antisense oligonucleotide (ASO) and siRNA techniques. Compared to siRNA therapies, such as inclisiran, ASOs are single stranded and independently target mRNA, without interacting with RISC [25]. In 2017, Graham et al. conducted a preclinical study and a phase 1 trial in healthy participants of an ASO targeting ANGPTL3 mRNA, ANGPTL3-L_RX_ [21]. In the preclinical study, the authors found lowering of LDL-C and triglycerides in all the mice treated with ANGPTL3 ASO therapy, as well as reduced hepatic triglyceride content [21]. Similar results were seen in the phase 1 trial, with maximal reductions in LDL-C, triglycerides, and apolipoprotein B by 46.5%, 51.7%, and 36.7%, respectively, compared to placebo, with no significant adverse events [21]. However, phase 2a and phase 2b trials had only modest LDL-C reductions, despite using higher doses of ANGPTL3 ASO therapy; although, the phase 2 study participants had lower baseline LDL-C values [21,59,60]. Additionally, in the phase 2 studies, there were concerning increases in liver enzymes and hepatic fat content, especially at higher doses, leading to its discontinuation in development [59,60,61].

The second approach to gene silencing ANGPTL3 is with ARO-ANG3, a siRNA-based therapy. In a phase 1 trial, 12 healthy volunteers received ARO-ANG3 at day 1 and a repeat dose at day 29 and had maximal reductions in LDL-C of 45–54% at around 5 weeks that was largely sustained to 16 weeks [24]. Similar preliminary results were seen in 17 patients with heterozygous familial hypercholesterolemia and an LDL-C ≥ 130 mg/dL despite maximally tolerated statin therapy, where after 16 weeks, LDL-C levels were reduced 23–37%, with no severe adverse events [62]. Further knowledge regarding ARO-ANG3 and safety information will be obtained from actively recruiting phase 2 trials (ClinicalTrials.gov Identified NCT05217667 and NCT04832971) [22,23].

## 7. Cholesteryl Ester Transfer Protein (CETP) Inhibitors

Observational studies have shown that loss of function of *CETP* is associated with an increase in high-density lipoprotein-cholesterol (HDL-C) and decrease in LDL-C [63]. CETP is involved in transferring triglycerides from atherogenic lipoprotein particles to HDL-C and cholesterol esters from HDL-C to atherogenic lipoproteins [63]. The original CETP inhibitors predominantly increased HDL-C, with minimal impact on LDL-C, but were terminated in development due to safety concerns and futility [63]. The newer CETP inhibitors, anacetrapib and evacetrapib, have been shown to not only increase HDL-C, but also decrease LDL-C [26]. In a phase 3 randomized control trial including 12,092 participants with elevated ASCVD risk, evacetrapib was found to increase HDL-C by 134.8% and decrease LDL-C by 37.1% compared to placebo [26]. Despite these favorable lipid profile results, treatment with evacetrapib did not result in lower cardiovascular risk, which is surprising as the LDL-C reduction achieved was similar to that of moderate intensity statin therapy [26]. A possible explanation for this negative result could be inadequate follow up time to demonstrate a cardiovascular benefit, as participants were only followed for slightly over 2 years [26]. Anacetrapib was also studied in a phase 3 trial that included adults over the age of 50 years with a history of ASCVD and was shown to increase HDL-C by 104% and reduce LDL-C by 41% compared to placebo [27]. Even more, after four years, the group treated with anacetrapib was found to have reduced cardiovascular risk (HR 0.91, 95% CI 0.85, 0.97) compared to the placebo group [27]. Although there were no major safety events, anacetrapib was found to persist in adipose tissue, providing concerns for prolonged duration of action [27]. Given the negative phase 3 trial for evacetrapib and concerns for drug accumulation in adipose tissue for anacetrapib, the development of both therapies has been discontinued [64].

Obicetrapib is the newest member of the CETP inhibitor class of medications [63]. Compared to the older CETP inhibitors, obicetrapib was created as a tetrahydroquinoline derivative, improving its oral absorption and potency [63]. In a phase 2 trial, 364 patients were randomized to obicetrapib or placebo, and after 12 weeks, LDL-C values were reduced by 26.6–44.5% compared to placebo [28]. Even more, combination therapy of obicetrapib and atorvastatin 20 mg led to an additional reduction in LDL-C by 50.2% compared to atorvastatin alone [28]. Currently, there is an actively recruiting randomized controlled cardiovascular outcomes trial, PREVAIL, studying if 10 mg of obicetrapib in secondary prevention lowers risk of major adverse cardiac events (ClinicalTrials.gov Identified NCT05202509) [29].

## 8. Discussion

An unmet need exists to appropriately lower LDL-C to reduce residual cardiovascular risk in primary and secondary prevention for individuals on maximally tolerated statin therapy who have not met LDL-C goals [1,2,4]. Despite the efficacy and safety of statin therapy, adherence rates remain low in primary and secondary prevention, at approximately 37% and 64%, respectively, with many not able to achieve LDL-C targets [65]. Currently, new therapies exist to lower residual cardiovascular risk in those with a history of ASCVD who have an LDL-C ≥ 70 mg/dL or severe primary hypercholesterolemia with an LDL-C ≥ 100 mg/dL while on the maximally tolerate statin therapy [7].

Ezetimibe is recommended for this patient population as a first-line add-on therapy given its low cost; although, its LDL-C lowering ability is modest at only 20% [1,4]. Thus, further LDL-C-lowering may be needed, which can be achieved with alirocumab and evolocumab; however, these monoclonal antibodies against PCSK9 are expensive and require injections every two weeks, limiting widespread use [45,46,47]. Recently, the FDA approved inclisiran, an siRNA therapy silencing PCSK9 expression capable of reducing LDL-C by 51%, requiring only twice-yearly injections [12,13]. Another medication that can be used in this population to lower residual risk is bempedoic acid, a prodrug targeting ATP-citrate lyase in the cholesterol synthesis pathway [1,4,10]. While not as effective as PCSK9 inhibitors in lowering LDL-C, bempedoic acid is cost-effective and oral; even more, the combination of bempedoic acid and ezetimibe lowers LDL-C by only a third less than PCSK9 inhibitors [40].

In addition to these new therapies to lower LDL-C, emerging therapies are underway that show promise if they further reduce residual ASCVD risk [18,19,24,62]. One promising avenue is with oral therapy targeting plasma PCSK9, MK-0616, and PCSK9 mRNA, AZD8233, capable of reducing LDL-C by 65% in a phase 1 study and 45–50% in cynomolgus monkeys, respectively [14,15,16,17,66]. Another target is ANGPTL3, which can be inhibited with fully human monoclonal antibodies or gene-silencing technology [19,24,62]. The monoclonal antibody against ANGPTL3, evinacumab, has been shown to reduce LDL-C by approximately 50% in individuals with homozygous familial hypercholesterolemia on maximally tolerated lipid-lowering therapy [19]. Additionally, ARO-ANG3, an siRNA targeting ANGPTL3, also shows promise, although more data are needed, which will be obtained in actively recruiting phase 2 trials [22,23]. Another emerging therapy is obicetrapib, the newest and most potent CETP inhibitor, shown to reduce LDL-C by 26.6–44.5% in a phase 2 study, with an actively recruiting cardiovascular outcomes trial underway [28]. Lastly, CRISPR adenine base editing therapy causes permanent loss of function of *PCSK9*, which has been shown to reduce LDL-C by 60% in non-human primates [18]. The advantages and disadvantages of these new and emerging therapies that have been FDA approved are seen in Table 2.

In conclusion, there has been tremendous innovation regarding targets and drug delivery techniques to lower LDL-C. Although statin therapy is the cornerstone to reducing cardiovascular disease risk in primary and secondary prevention, several individuals require additional LDL-C-lowering therapy. These new and emerging therapies show promise in reducing residual cardiovascular disease risk.

## Figures and Tables

**Table 1 jcm-11-04611-t001:** Current, new, and emerging LDL-C lowering therapies.

Drug	Target	LDL-C Impact	Clinical Use or Status
Statins [7,8]	(HMG)-CoA reductase	~50%	DM Severe hypercholesterolemia ASCVD PCE ≥ 7.5%
Ezetimibe [7]	NPC1L1	~20%	Add-on to statin therapy for: ASCVD and LDL-C ≥ 70 mg/dL Severe hypercholesterolemia and LDL-C ≥ 100 mg/dL
Bempedoic acid [9,10,11]	ATP-citrate lyase	~20–25%	Add-on therapy (FDA 2/2020): ASCVD Heterozygous familial hypercholesterolemia
Alirocumab and Evolocumab [7]	Plasma PCSK9	~60%	Add-on to statin and ezetimibe therapy (FDA 7/2015): ASCVD and LDL-C ≥ 70 mg/dL Severe hypercholesterolemia and LDL-C ≥ 100 mg/dL
Inclisiran [12,13]	PCSK9 mRNA	~50%	Add-on therapy (FDA 12/2021): ASCVD Heterozygous familial hypercholesterolemia
MK-0616 [14,15,16]	Plasma PCSK9	~65%	Phase 2 upcoming
AZD8233 [17]	PCSK9 mRNA	~45–50%	Non-human primate data
VERVE-101 [18]	*PCSK9*	~60%	Non-human primate data
Evinacumab [19,20]	ANGPTL3	~50%	Add-on therapy (FDA 2/2021): Homozygous familial hypercholesterolemia
ANGPTL3-L_RX_ [21]	ANGPTL3 mRNA	~50%	Development Terminated
ARO-ANG3 [22,23,24,25]	ANGPTL3 mRNA	~50%	Phase 2, actively recruiting
Evacetrapib [26]	CETP	~40%	Development Terminated
Anacetrapib [27]	CETP	~40%	Development Terminated
Obicetrapib [28,29]	CETP	~45%	Phase 3, actively recruiting

HMG-CoA = β-hydroxy β-methylglutaryl-coenzyme A, DM = diabetes mellitus; ASCVD = atherosclerotic cardiovascular disease; PCE = pooled cohort equations; NPC1L1 = Niemann-Pick C1-Like 1; LDL-C = low-density lipoprotein-cholesterol; PCSK9 = Proprotein convertase subtilisin-kexin type 9, ANGPTL3 = Angiopoietin-related protein 3; CETP = Cholesteryl ester transfer protein.

**Table 2 jcm-11-04611-t002:** Advantages and disadvantages of LDL-C-lowering therapies.

Drug	Advantages	Disadvantages
Statins [7,8]	Excellent LDL-C lowering Strong evidence reducing ASCVD risk Six of the seven statins are generic	Low adherence
Ezetimibe [7]	Well tolerated Moderate evidence in secondary prevention	Modest LDL-C lowering
Bempedoic acid [9,10,11]	Well tolerated in those with statin associated side effects	Modest LDL-C lowering
Alirocumab and Evolocumab [7]	Excellent LDL-C lowering Well tolerated Strong evidence reducing ASCVD risk	Cost Injection
Inclisiran [12,13]	Excellent LDL-C lowering Durability (only 3 injections first year)	No ASCVD outcome data
Evinacumab [19,20]	Excellent LDL-C lowering (Use in homozygous familial hypercholesterolemia)	Cost Injection

LDL-C = low-density lipoprotein-cholesterol; ASCVD = atherosclerotic cardiovascular disease.

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
