# Peer review of "Newer and Emerging LDL-C Lowering Agents and Implications for ASCVD Residual Risk"

_jcm, 2022, doi:10.3390/jcm11154611_

Round 1

Reviewer 1 Report

In this article, in addition to statin, ezetimibe and PCSK9 inhibitor, the authors made a review about the other emerging therapies for lowering low-density lipoprotein-cholesterol (LDL-C), including bempedoic acid, angiopoietin-related protein 3 (ANGPTL3) and cholesteryl ester transfer protein (CETP) inhibitors. The results of clinical trials and the future development of these emerging therapies were described. Basically, this is a well-written review article that provides comprehensive information about current therapies for LDL-C reduction. Only several minor corrections are necessary.

1.     Lin 98 and line 139. The sequence was reversed. The section 3 is ezetimibe, section 5 is bempedoic acid. But section 4 PCSK9 inhibitor was after section 5.

2.     Line 139. It should be “Proprotein convertase subtilisin-kexin type 9 (PCSK9) INHIBITOR”.

3.     Line 324. In addition to monthly injection, the more commonly used dosage of PCSK9 inhibitors also include every 2 weeks injection.

Reviewer 2 Report

1. I suggest the authors to briefly mention the new and emerging approaches/agents/therapies in the Abstract section. It may attract more attention or broaden the readership including patients.

2.     I suggest to add the information on the approved date by FDA and on the drug administration route into Table 1.

3.     It will be better to concisely show the advantages and disadvantages of each of the new and emerging agents/therapies in a new table (i.e., Table 2).

4.     Typos: “5. Bempedoic acid” should be “4. Bempedoic acid” whereas “4. PCSK9” should be “5. PCSK9”.
